# A Comparative Review on Greenery Ecosystems and Their Impacts on Sustainability of Building Environment

**Hussain H. Al-Kayiem** [1,*] , **Kelly Koh** [1] , **Tri W. B. Riyadi** [2] **and Marwan Effendy** [2]

[1] Mechanical Engineering Department, Universiti Teknologi PETRONAS, Seri Iskandar 32610, Malaysia; kelly_16005693@utp.edu.my

[2] Mechanical Engineering Department, Universitas Muhammadiyah Surakarta, Surakarta 57102, Indonesia; Tri.Riyadi@ums.ac.id (T.W.B.R.); Marwan.Effendy@ums.ac.id (M.E.)

\* Correspondence: hussain_kayiem@utp.edu.my; Tel.: +60-5-368-7008

**Abstract:** Greenery systems are sustainable ecosystems for buildings. Many studies on greenery systems, such as green roofs and green walls, have demonstrated that greenery systems support energy saving and improve thermal conditions in the building sector. This paper summarizes, discusses, and compares greenery systems and their contributions to the reduction of the urban heat index, the reduction of internal and external buildings' wall temperatures, and the reduction of the energy consumption of buildings. The fundamental mechanisms of greenery systems, which are thermal insulation, evapotranspiration, and shading effect, are also discussed. The benefits of greenery systems include the improvement of stormwater management, the improvement of air quality, the reduction of sound pollution, the reduction of carbon dioxide, and the improvement of aesthetic building value. The summarized materials on the greenery systems in the article will be a point of references for the researchers, planners, and developers of urban and rural areas, as well as the individual's interest for future urban and rural plans.

**Keywords:** building environment; ecosystem; energy saving; greenery systems; living green wall; urban heat index; sustainable buildings

---

## 1. Introduction

Based on the United Nations Population Division, the worldwide population is drastically increasing at about 80 million annually or 1.1 percent per year [1]. Urbanization refers to the physical population growth of rural areas, horizontally and vertically. In the process of urbanization, green lands are turning into concrete jungles, typically having a high building density. The growth of population and urbanization are often associated with an upward trend of energy demand and natural resources, such as fossil fuel, which burden the ecosystem [2]. Urbanization also raises the problem of more energy demand, as consumption is increasing.

Energy consumption is distributed among four main sectors, which include industrial, building (commercial and residential), transportation, and agriculture. The International Energy Agency predicts that the global population will grow by 2.5 billion by 2050 and the energy demand in the building sector will also increase sharply by 50%. The building sector accounts for a large proportion of primary energy consumption. In developed countries, the building sector consumes between 20 percent and 40 percent of the total energy of the building [3,4]. In China, buildings contribute to approximately 28 percent of energy consumption [5] and buildings in United States contribute to 40 percent of energy consumption [6]. The percentages of the energy consumed in the building sector for

a few selected countries are shown in Table 1 below. Meanwhile, in Malaysia, the energy consumption of the building is at about 53.6 percent of the total energy consumption and 14.6 percent of the final energy demand [7].

**Table 1.** Percentage of energy consumption in the building sector for some selected countries [8].

| Country | Percentage (%) |
|---|---|
| United States of America | 18 |
| Hong Kong | 30 |
| Japan | 26 |
| China | 35 |
| Thailand | 33 |

Indoor thermal comfort is greatly dependent on the operations of heating, ventilation, and air conditioning systems (HVAC). Along with climate change, the operation of HVAC systems, and heating and cooling, energy load also increases tremendously [9]. As the building sector has dominated the total energy consumption, the efficient use of energy in buildings is one of the most cost-effective measures to reduce the environmental impact. Considering that they are the primary source of energy production, fossil fuels, such as natural gas and coal, are in limited supply, thus placing additional pressure on the energy system. Among other innovative technologies to improve the thermal performance of buildings are greenery systems, such as green roofs and green walls, which display significant energy reduction and facilitate urban adaption to a warming climate [10].

This review paper aims to discuss the literature on greenery systems and the contribution to the mitigation of urban heat islands, the reduction of energy consumption, and the reduction of internal and external wall temperatures. The paper is divided into seven main sections. In Section 1 of this paper, the definition and classification of urban heat island is reviewed. The mitigation techniques that are influencing the urban heat index may be the greenery systems applied on buildings, or urban green spaces that include large land and large scale systems, such as lakes and parks. As the theme of the current review is comparative on the greenery system influencing the interior environment, thermal comfort, and cooling load on buildings, no emphasis has been placed on urban green spaces and green open space systems. In Section 2, the definition and classification of greenery systems is reviewed. The fundamental mechanism of greenery systems, which contributes to the thermal performance and energy saving in buildings, are discussed in Section 3 of this paper. Additionally, the environmental benefits of the greenery systems will be discussed in Section 4. As different types of plant species have different characteristics, it will affect the performance of the greenery systems. Moreover, the growth of plant species varies in different climates. Thus, the selection of plant species and climate is reviewed in the following section. Previous studies have suggested that greenery systems demonstrate positive results in terms of the mitigation of urban heat islands, energy saving and the reduction of energy consumption, and the reduction of internal and external wall temperatures. Thus, several papers are reviewed and summarized in the last section of this paper to provide evidence.

## 2. Urban Heat Island (UHI)

The process of urbanization is converting green lands into a concrete jungle with a high density of buildings. Urbanization results in the replacement of greenery with urban fabric and causes a significant change in the properties of the land surface by modifying the surface energy balance of the urban area [11]. In other words, urbanization promotes the change of the land profile on Earth. The urban building has greater thermal properties, which results in higher temperatures in the urban area compared to the surrounding rural area [12]. The maximum temperature difference between the urban area and rural area is the effect of urban heat islands [13,14], as described in Figure 1 below. Urban heat islands (UHIs) contribute to thermal discomfort and higher energy loads in mid and

low latitude countries, whereas it can function as an asset in reducing heating loads in high latitude countries with cooler climates [15].

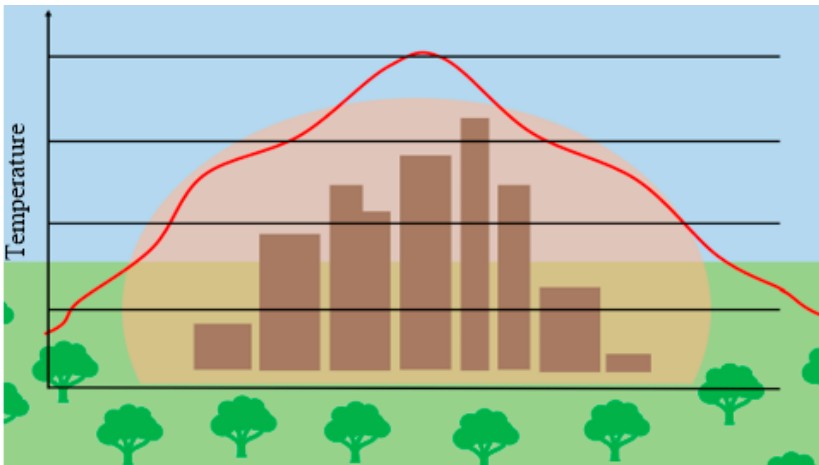

**Figure 1.** Illustration of urban heat island.

The modification of land surface alters the thermal properties, causing the urban areas to store more radiated heat or sensible heat inside the space during the day and release the heat back into the atmosphere at night. In the urban area, the heat transfer is minimal due to humidity in the atmosphere, which creates a hostile condition for the inhabitants of the area [16]. The changing material results in the new surface and atmospheric conditions, thereby altering the exchange of energy and airflow.

In accordance with the spatial coverage and temporal development, UHIs are divided into two general classifications: urban canopy layer and urban boundary layer. An urban canopy layer has limited coverage of a local scale between 1 to 10 km, where the layer is below the top of the roof and tree level [17]. This phenomenon only occurs during the night, as it affects the thermal comfort and urban ecology. The urban canopy layer contributes to the heating, cooling, and evaporation of the urban boundary layer [18]. On the other hand, the urban boundary layer has a high coverage of eco scale at 10 km above ground from the mean roof or tree level and extends up to the urban landscape [17]. This phenomenon occurs during both day and night and affects the local air circulation. The urban boundary layer contributes to a greater mesoscale weather condition [18].

The urban area replaces Earth's natural greenery of near-uniform surface roughness with canyon geometry, where the buildings are very tall and dense. Urban buildings are usually dark surface materials, non-reflective, and impermeable, such as asphalts, concrete, brick, metal, and glass [19]. In comparison with the natural greenery, these materials have greater thermal properties. During the daytime, the short-wave and long-wave radiation or sensible heat will be trapped in the canyon geometry and is stored inside the building before being released into the atmosphere at night [20]. On the other hand, the layer of greenery has a cooling effect by converting the incident energy into latent heat instead of sensible heat through the process of evapotranspiration [21]. Hence, the loss of natural greenery exacerbates the UHI effect. In addition to this, anthropogenic heat sources caused by human activities, such as vehicle combustion, industrial combustion, and air conditioning systems, also aggravate the UHI effect [22].

## 3. Greenery Systems: Definition and Classification

The energy efficiency of the building is dependent on the building skin responding to air conditioning and artificial lighting needs. The building skin plays an important role on the total energy consumption of the building by controlling the transfer of thermal heat into the building [23,24]. Greenery systems are one of the approaches among other innovative technologies to improve the thermal performance of a building, as it displays significant energy reduction and facilitates urban

adaption to a warming climate [10]. There are three different classifications of the greenery system, which are the green roof, green façade, and living wall, as shown in Figure 2.

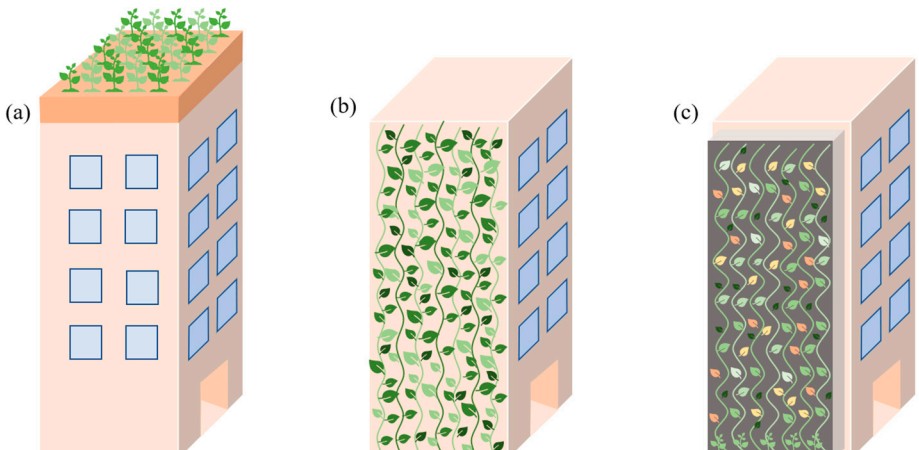

(a)  (b)  (c)

**Figure 2.** Illustration of greenery systems: (**a**) green roof, (**b**) green façade, and (**c**) living wall.

## 3.1. Green Roof

The roof of a building is approximately 20 percent of the total building structure surface [25]. Thus, providing a green layer on the roof will display a significant influence on the thermal performance of the building. A green roof is defined as the utilization of different supportive layers, which provides a suitable condition for the plants to form a green landscape on the roof [26–29]. Research suggests that the green roof can reflect 27 percent of solar radiation, absorb 60 percent of solar radiation through the process of photosynthesis, and transmit 13 percent to the growing medium [30].

The composition of a green roof is shown in Figure 3 below. The supportive layer includes: waterproof membrane—a layer of membrane which protects the root from decaying; filter membrane—a layer of membrane which prevents fine residue from infiltrating into the drainage layer; drainage layer—a layer to remove excess water to prevent water clogging in the system; growing medium—a layer of material composed of inorganic and organic matter, which provides a suitable growing medium to grow plants; and plants—a layer of green vegetation [31,32].

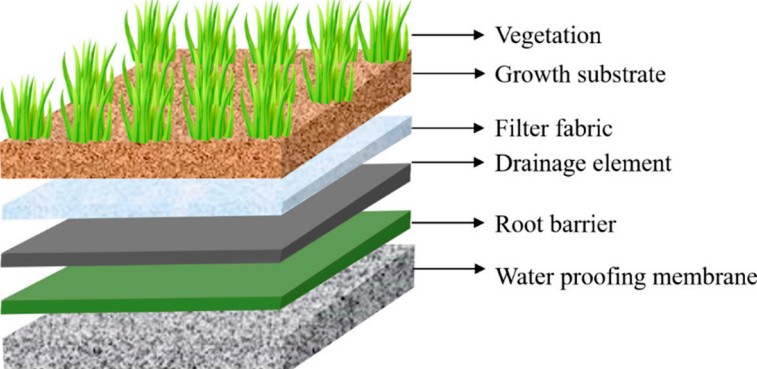

→ Vegetation
→ Growth substrate
→ Filter fabric
→ Drainage element
→ Root barrier
→ Water proofing membrane

**Figure 3.** Composition of a green roof.

In accordance with the type of usage, construction factors, and maintenance requirements, the green roof is divided into two classifications: intensive and extensive. An intensive green roof has similar management to a ground level garden to provide amenity space and is accessible, as shown in Figure 4 below. It has a thick growing medium of more than 15 cm up to 200 cm, which requires higher construction cost and maintenance. Due to the increased soil depth, an intensive green roof has higher

weight and has a wide variety of plants, such as shrubs and small trees [26,31,33,34]. The variety of plants creates an appealing natural environment with improved biodiversity [35].

On the other hand, an extensive green roof has lower management requirements and is not publicly accessible, as shown in Figure 5 below. It has a thin growing medium at about less than 15 cm, which contributes to lower construction cost and maintenance. The extensive green roof is lightweight and can only accommodate a limited variety of plants, including grasses and moss [26,31,33,34]. When comparing the two types of green roof, the extensive green roof is a more common option considering the weight restrictions, where certain roofs cannot tolerate unexpected loads, lower construction cost, and maintenance [33,36]. In addition, the extensive green roof is also suitable for a large sized rooftop, where the construction process is technically simple and allows implementation on a sloped roof [35].

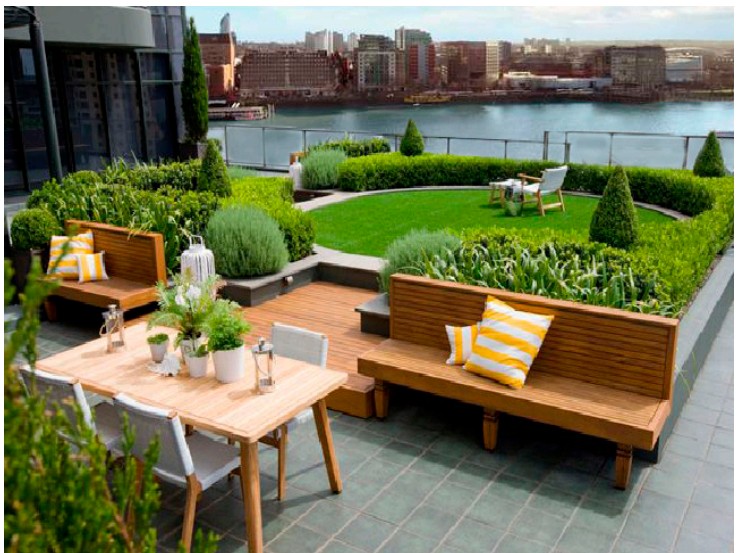

**Figure 4.** Intensive green roof [37].

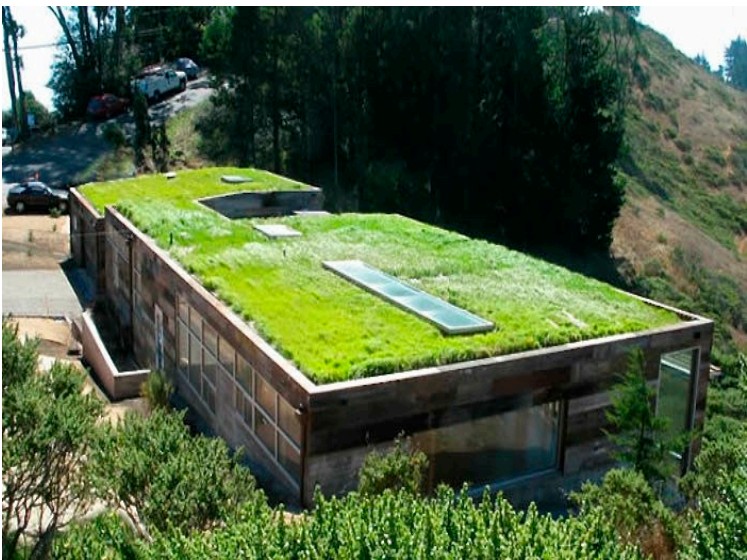

**Figure 5.** Extensive green roof [38].

*3.2. Green Wall*

The walls of a building occupy a high fraction of the total building structure surface, with the total wall areas potentially greater than the space compared roof [39]. In case of a high-rise building, the surface area of a wall is 20 times greater than the roof [40]. A green wall has greater potential

compared to a green roof considering that a green wall can double the ground footprint of buildings [41]. The definition of a green wall is climbing plants grown in a supported vertical system either directly against or on supported structures integrated into external building walls [42]. The green wall is divided into two general classifications: green façade and living wall, as shown in Figure 6.

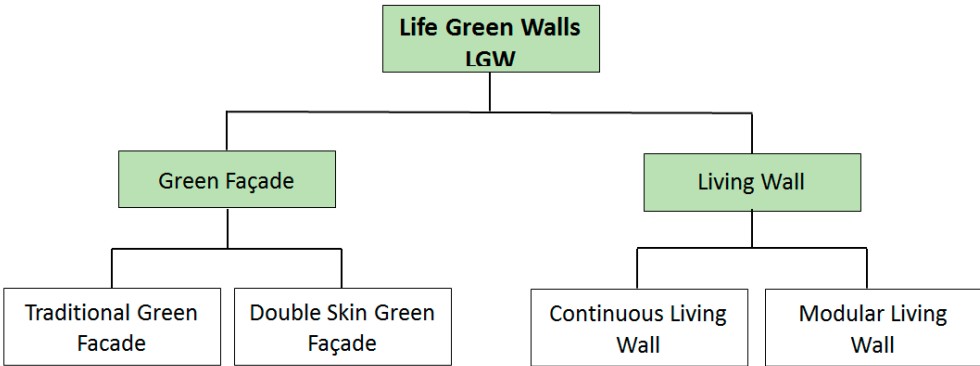

**Figure 6.** Classification of green wall.

*3.3. Green Façade*

A green façade is divided into two classifications: traditional green façade and double skin green façade. Figure 7 illustrates the difference between the traditional green façade and the double skin green façade.

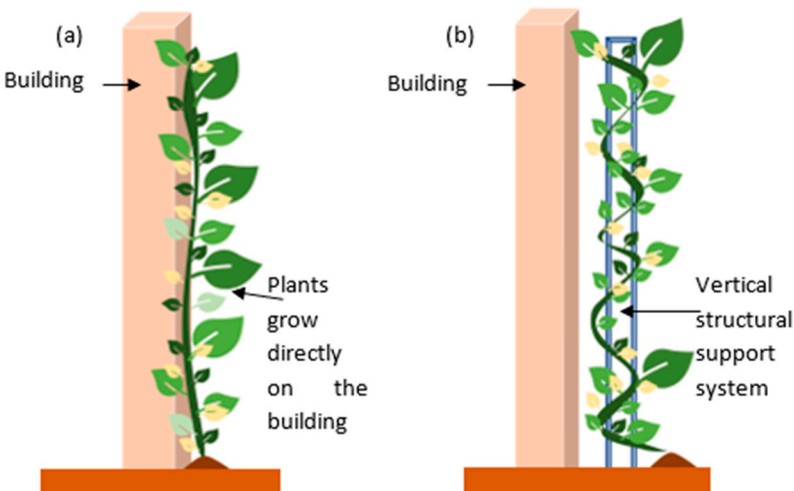

**Figure 7.** Classification of green façade: (**a**) traditional green façade, (**b**) double skin green façade.

In a traditional green façade, deciduous climbing plants are rooted in the ground and use the building envelope to cover the wall of the building, as shown in Figure 8 below [43]. As the climbing plants use the building envelope as the structure, there is a risk of damage to the wall of the building [42]. In addition to that, when the climbing plants have a full coverage of the wall, there is a risk of the greenery layer falling due to heavy weight. However, a traditional green façade is the most cost-effective among other greenery system methods.

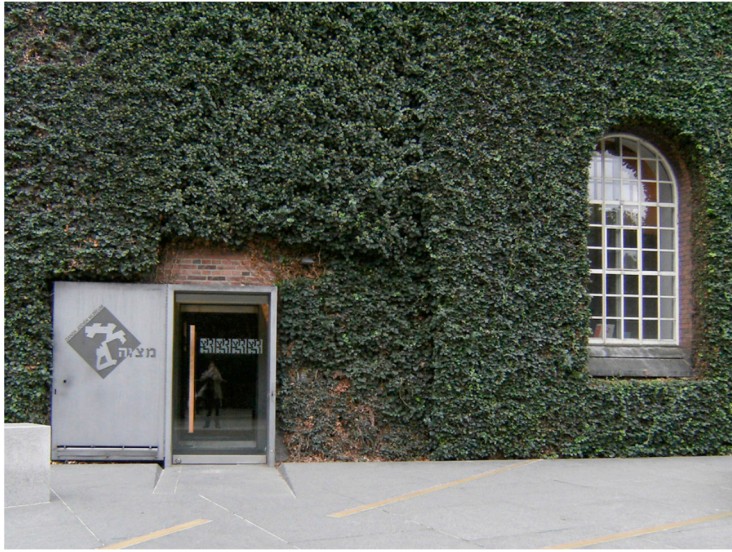

**Figure 8.** Traditional green façade [44].

On the other hand, a double skin green façade requires a vertical structural support, such as modular trellis, stainless steel cables, or stainless steel mesh to guide the plant's climb along the wall of the building like a second layer of skin, as shown in Figure 9 below [42,44,45]. In a double skin green façade, the framework is installed at a distance from the wall, creating a gap or cavity between the wall of the building and the plants. The distance of the cavity influences the rate of air exchange, which affects the wall surface temperature and indoor air temperature [46]. As the distance of the gap increases, the temperature inside the cavity decreases, resulting in higher wall surface temperature and indoor air temperature. According to the research conducted, the optimum distance of the gap is 30 cm [47].

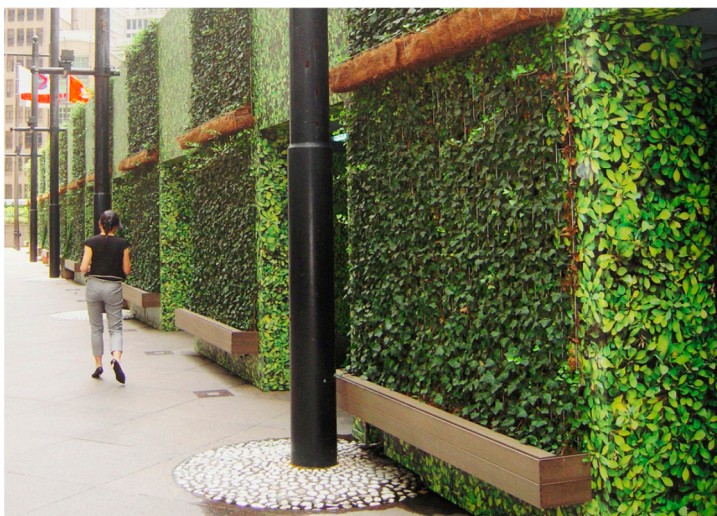

**Figure 9.** Double skin green façade [44].

### 3.4. Living Wall

In comparison to the green façade, a living wall has a more complex structure, including special supporting elements, growing media, and an irrigation system to serve a large diversity of plants [48]. The living wall is involved in the recent innovation of wall cladding and is as displayed in Figure 10 below. The living wall is composed with pre-vegetated panels that are fixed to a structural wall of a free-standing frame to allow a rapid coverage of large surfaces and a more uniform growth along

the wall of the building, thus reaching the top of high buildings [44,49,50]. The living wall allows the development of the aesthetic concept of the green wall based on the variation of plant color and density [44]. Hence, a living wall system has a high construction cost due to its complexity to provide a variety of plant options with fast and good coverage on a very tall building.

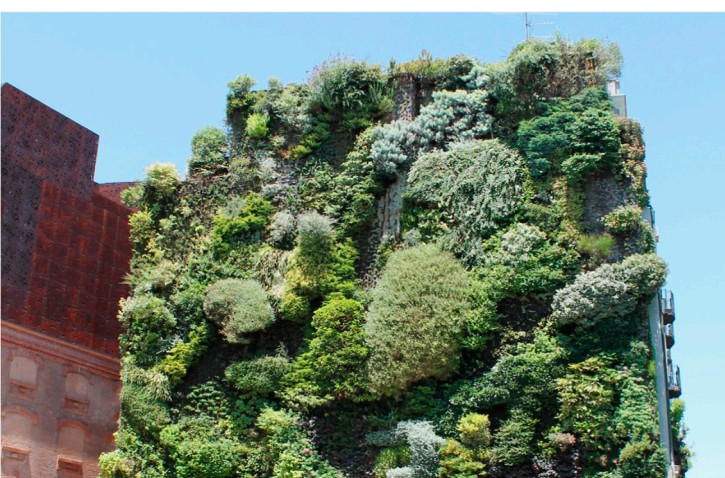

**Figure 10.** Living wall [44].

The living wall is divided into two classifications: continuous and modular. The continuous living wall is based on the installation of a frame fixed to the wall, forming a void space between the system and the surface. This frame holds the base of the panels and protects the wall from humidity. Meanwhile, the base panels support the permeable fabric layer, such as geotextile membrane [44]. The continuous living wall is very lightweight as it does not require a growing medium, where plants grow using hydroponic techniques [29]. However, continuous living walls require a constant irrigation system to provide necessary nutrients to the plants.

On the other hand, a modular living wall is composed of several interlocked parts, which include trays, vessels, planters, tiles, or flexible bags. The growing medium in a modular living wall is usually a mixture of a light substrate with a granular material in order to provide a good water retention capacity [42]. Each of the modular living wall components is designed to hold the growing medium and is fixed to the supporting structure, providing an advantage of extra planting depth and easy maintenance [51].

## 4. Fundamental Mechanism of Greenery Systems

As the Earth's natural greenery is replaced with urban fabric, many have suggested that greenery systems, such as a green roof, green façade, and living wall, have a great impact in mitigating energy demand by enhancing the thermal performance of the buildings. The thermal performance of the building depends on the structural details of the building envelope. Urban fabrics, such as asphalts, concrete, brick, metal, and glass, have higher thermal properties compared to a layer of green vegetation. Regarding the operation, the greenery systems act through a few fundamental mechanisms: thermal insulation, evapotranspiration, and shading effect, which will be discussed in this section.

Thermal insulation is greatly affected by the properties of different layers in the greenery systems. In the case of the green roof, the growing medium provides an extra layer of insulation to the roof [33]. A substrate with less density has higher porosity, providing additional air pockets to improve the thermal insulation properties of the growing medium [52]. In addition to that, the presence of moisture in the growing medium also influences the thermal properties of the green roof. During the summer season, a growing medium with more water content provides additional evapotranspiration by dissipating heat from the building [26]. As water has higher thermal conductivity than air, a growing medium with less water content improves the thermal performance of a green roof during winter [53,54].

In simpler terms, a wet growing medium enhances the conduction and convection during summer and a dry substrate increases the heat storage during winter [28].

In case of a double skin green façade, a layer of stagnant air in the gap between the layer of plants and building envelope acts as a thermal buffer that serves as an extra layer of thermal insulation [55]. Previous research suggests that providing a layer of insulation on the external building envelope is more effective during the summer season [56]. The rate of air exchange in the gap influences the convective heat transfer on the external wall surface of the building, resulting in the reduction of heat flux through the building envelope [57]. In other words, the gap in the double skin green façade regulates the ambient temperature and wind speed around the building.

Evapotranspiration is a combination of two phenomena: evaporation and transpiration. Evapotranspiration describes the water consumed by plants over a period of time. After the period, heat energy is absorbed by the process of evapotranspiration, where the plants lose water during the process of evaporation and transpiration. Evaporation is a physical process where water evaporates from the soil into the atmosphere. Meanwhile, transpiration is a physiological process where water is lost through the stomata of the leaves into the atmosphere [58]. The stomata of the leaf are intercellular openings between the epidermal cell of the leaf surface [35]. Hence, the rate of transpiration is determined by the stomatal resistance [59]. Evapotranspiration is the main factor contributing to the cooling effect of the building, as plants can dissipate solar radiation. A large amount of solar radiation is converted into latent heat during the process of evapotranspiration, resulting in the reduction of temperature on the surface of the building wall and room temperature [60]. The rate of evapotranspiration is greatly influenced by humidity, growing medium, wind speed, type of plants, and local climate consisting of the solar radiation and temperature [61].

The thermal performance of a greenery system is also influenced by the shading effect. The amount of solar radiation is absorbed, reflected, and transmitted [62]. The ratios of solar radiation absorbed, reflected, and transmitted vary according to the type of plant [51]. Leaf area index and leaf angle distribution are the important parameters affecting the shading effect in a greenery system. Leaf area index is the representation of the area coverage of the leaves. It describes the relationship between the area and the area of the floor [63]. It is a dimensionless value between zero and ten to define the characteristics of a green layer [45]. As the value of the leaf area index increases, the solar radiation cannot transmit into the surface of the building, providing a great shading effect for the building [64]. In the case of vertical greenery systems, the leaves respond to the high-angled sun, resulting in a ventilation blind effect, where warm air escapes and is replaced by cooler air [65].

## 5. Benefits of Greenery Systems

To address the matter regarding the depletion of renewable resources for the generation of electricity, which results in serious environmental impacts, such as an urban heat island, greenery systems are an effective approach combining nature and buildings. Several studies have been conducted on the environmental benefits of greenery systems, which include improving the stormwater management, improving air quality, the reduction of sound pollution, the reduction of carbon dioxide, and the improvement of the aesthetic value of the building.

The installation of a green roof can improve the stormwater management by retaining rainwater and delaying the peak flow, thereby reducing the risk of flood [66,67]. The rainwater is either absorbed by the vegetation, growing medium, or drainage element in the green roof [33]. Through the process of evapotranspiration, the rainwater absorbed by the vegetation will be stored in the stomata to be transpired and rainwater in the growing medium will evaporate [68].

Plants in the greenery systems improve the air quality through direct and indirect processes. The plants directly consume gaseous pollutants through their stomata or indirectly modify the microclimates [33]. Leaves in the greenery systems filter airborne particles and contaminate. Meanwhile, the branches absorb noxious gases through photosynthesis processes [21]. In past research, the urban forest model had the potential of a green roof for air pollution removal. An urban forest effect model

was carried out in Chicago and Detroit, where 109 hectares of green roof contributed to 7.87 metric tons of air pollution removal per year [35].

The greenery systems also act as a barrier against urban noise pollution. The effect of greenery systems on the acoustics of the building were studied experimentally by [24]. They concluded that the green roof and green wall had an acoustic insulation at about 10 decibels and 30 decibels, respectively, compared to an exposed roof and wall. The reduction of sound in the green roof occurs due to diffraction; meanwhile, in the green wall, it is due to absorption of sound frequencies. There are a few factors affecting the degree of sound insulation, which include the depth of the growing medium and the type of plants and materials used for structural components.

Photosynthesis is a process occurring in plants, where light energy is captured by chlorophyll to break down carbon dioxide and water to be converted into other chemical components and oxygen. In other words, greenery systems have profound impacts on the reduction of carbon dioxide. The amount of carbon dioxide used by green plants varies according to the time of the day, where it increases from the morning and is at the peak in the afternoon but will decrease rapidly at night. Thus, plants can be defined as a carbon sink instead of a carbon source [29].

There are various studies suggesting that greenery systems improve the aesthetic value of the building by creating visual interest to hide unsightly features. In addition to that, it also increases the property value of the building.

## 6. Selection of Plants and Climatic Influence

A careful selection of plants in regard to the climatic condition, building characteristics, and type of system configuration is very important. Selecting an appropriate type of plant will greatly affect the performance of the system because different types of plants have different characteristics, including plant trait, leaf area index, foliage height, albedo, and stomatal resistance [32,69,70]. In addition to that, the selection of plants also depends on a few factors, such as preferred visual effect, availability of plant species, and requirement of an irrigation system [71].

In the case of a green roof, sedum is the most common type of plant used, mainly in an extensive green roof. Sedum grows across the ground, offering a good coverage [36]. Moreover, sedum is a succulent plant and is compatible with limited water sources, as it stores water in the stomata of the leaves [72]. Although it can provide a high shading effect against solar radiation, sedum has a low thermal resistance value, as it is unable to avoid convective heat transfer [73,74].

In the case of a green façade, climbing plants grow on the building envelope or supporting structure. There are two types of climbing plants: evergreen and deciduous. Evergreen plants maintain their leaves all year long; meanwhile, deciduous plants lose their leaves due to pruning during the fall season [44]. Blue trumpet vine is the most common type of plant used in a green façade. A blue trumpet vine has the ability to grow very fast in a limited time and offer a good coverage. In addition to that, it creates a consistent and adequate density with minimum pruning, resulting in suitability in a tropical climate, which is hot and humid all year long [75].

As discussed in the previous section, plants with a high leaf area index contribute to the better thermal performance of greenery systems. A high leaf area index value has a higher shading effect, resulting in better heat transfer into the building. Moreover, plants with light colored leaves and short foliage height have better cooling effects for the building [71]. Taking the climate and system configuration into consideration, mixing different species of plants will provide a denser canopy in a short period of time because different plants have different shapes and sizes, resulting in the covering of the gap. In addition to this, plants that are fast growing, have a high tolerance to adverse weather, and are low maintenance are preferable. However, experimental work must be conducted in different parts of the world, as different types of plants have different performances in different types of weather.

As the development of different plant species varies from one region to another, the weather condition over the operation is very important [76]. The climate greatly influences the growth of the plant and their physiological processes, resulting in an influence on the thermal performance of

the system configuration and building [77]. The Köopen climate classification is the most widely used to classify the world's climate [78]. The world climate classification is divided into five main groups of climate region and subcategories based on the annual and monthly precipitation and average temperature. The climate classification is summarized as shown in Figure 11 and Table 2. The types of plant species and climate classifications for respective greenery system configuration are also included in the next section to study the previous findings that were conducted.

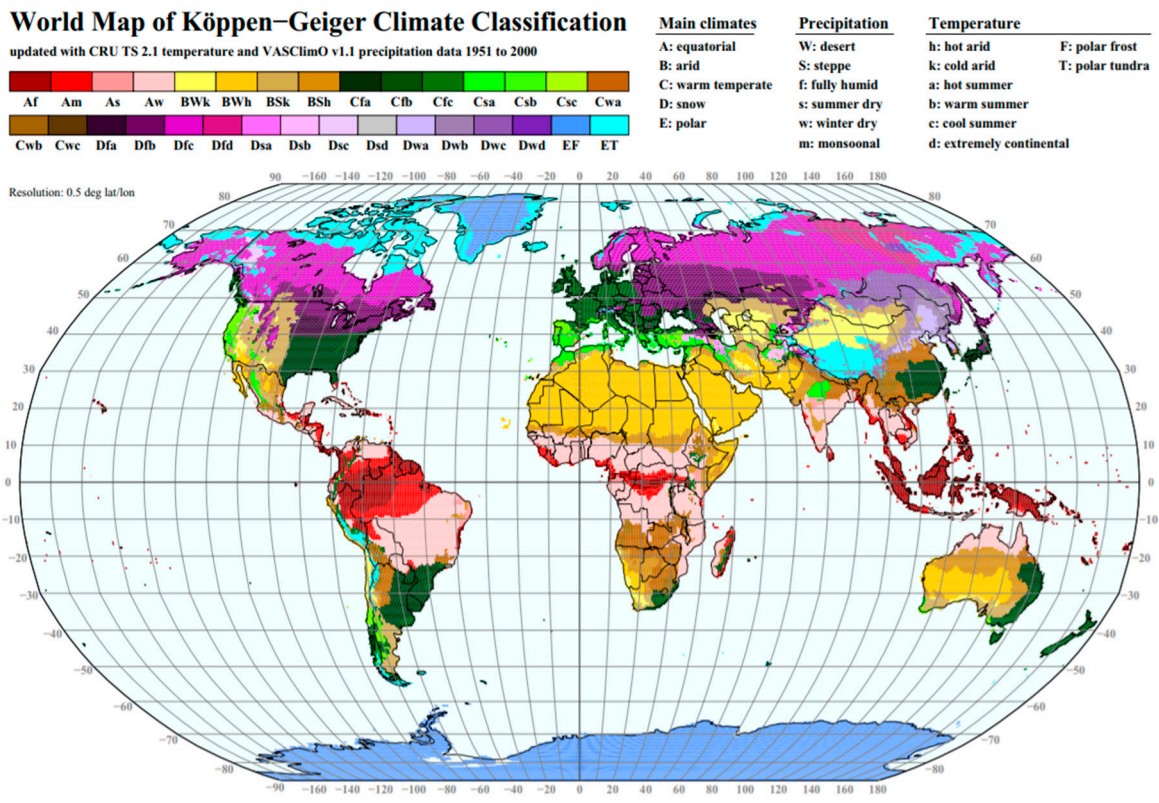

**Figure 11.** World map of Köopen–Geiger climate classification [78].

**Table 2.** Köopen–Geiger climate classification characteristics.

| Group | Köopen–Geiger Subcategories | | | Characteristic |
|---|---|---|---|---|
| Tropical | Af | (A) (f) | Equatorial Fully humid | Tropical rainforest climate |
| | Am | (A) (m) | Equatorial Monsoonal | Tropical monsoon climate |
| | Aw | (A) (w) | Equatorial Winter dry | Tropical wet and savanna climate |
| | As | (A) (s) | Equatorial Summer dry | Tropical dry and savanna climate |
| Arid | BWh | (B) (W) (h) | Arid Desert Hot arid | Hot desert climate |
| | BWk | (B) (W) (k) | Arid Desert Cold arid | Cold desert climate |

**Table 2.** *Cont.*

| Group | Köopen–Geiger Subcategories | | | Characteristic |
|---|---|---|---|---|
| | BSh | (B)<br>(S) | Arid<br>Steppe | Hot semi-arid climate |
| | | (h) | Hot arid | |
| | BSk | (B)<br>(S)<br>(k) | Arid<br>Steppe<br>Cold arid | Cold semi-arid climate |
| Subtropical | Csa | (C)<br>(s)<br>(a) | Warm temperate<br>Summer dry<br>Hot summer | Hot summer Mediterranean climate |
| | Csb | (C)<br>(s)<br>(b) | Warm temperate<br>Summer dry<br>Warm summer | Warm summer Mediterranean climate |
| | Cwa | (C)<br>(w)<br>(a) | Warm temperate<br>Winter dry<br>Hot summer | Monsoon-influenced humid subtropical climate |
| | Cwb | (C)<br>(w)<br>(b) | Warm temperate<br>Winter dry<br>Warm summer | Subtropical highland climate |
| | Cwc | (C)<br>(w)<br>(c) | Warm temperate<br>Winter dry<br>Cool summer | Cold subtropical highland climate |
| | Cfa | (C)<br>(f)<br>(a) | Warm temperate<br>Fully humid<br>Hot summer | Humid subtropical climate |
| | Cfb | (C)<br>(f)<br>(b) | Warm temperate<br>Fully humid<br>Warm summer | Temperate oceanic climate |
| | Cfc | (C)<br>(f)<br>(c) | Warm temperate<br>Fully humid<br>Cool summer | Subpolar oceanic climate |
| Continental | Dsa | (D)<br>(s)<br>(a) | Snow<br>Summer dry<br>Hot summer | Mediterranean-influenced hot summer humid continental climate |
| | Dsb | (D)<br>(s)<br>(b) | Snow<br>Summer dry<br>Warm summer | Mediterranean-influenced warm summer humid continental climate |
| | Dsc | (D)<br>(s)<br>(c) | Snow<br>Summer dry<br>Cool summer | Subarctic climate |
| | Dsd | (D)<br>(s)<br>(d) | Snow<br>Summer dry<br>Extremely continental | Extremely cold subarctic climate |
| | Dwa | (D)<br>(w)<br>(a) | Snow<br>Winter dry<br>Hot summer | Monsoon-influenced hot summer humid continental climate |

**Table 2.** *Cont.*

| Group | Köopen–Geiger Subcategories | | | Characteristic |
|---|---|---|---|---|
| | Dwb | (D)<br>(w)<br>(b) | Snow<br>Winter dry<br>Warm summer | Monsoon-influenced warm summer humid continental climate |
| | Dwc | (D)<br>(w)<br>(c) | Snow<br>Winter dry<br>Cool summer | Monsoon-influenced subarctic climate |
| | Dwd | (D)<br>(w)<br>(d) | Snow<br>Winter dry<br>Extremely continental | Monsoon-influenced extremely cold subarctic climate |
| | Dfa | (D)<br>(f)<br>(a) | Snow<br>Fully humid<br>Hot summer | Hot summer humid continental climate |
| | Dfb | (D)<br>(f)<br>(b) | Snow<br>Fully humid<br>Warm summer | Warm summer humid continental climate |
| | Dfc | (D)<br>(f)<br>(c) | Snow<br>Fully humid<br>Cool summer | Subarctic climate |
| | Dfd | (D)<br>(f)<br>(d) | Snow<br>Fully humid<br>Extremely continental | Extremely cold subarctic climate |
| Polar | ET | (E)<br>(T) | Polar<br>Polar tundra | Tundra climate |
| | EF | (E)<br>(F) | Polar<br>Polar frost | Ice cap climate |

## 7. Discussions on Greenery System's Influence on Building Environment

A considerable amount of research has been conducted to study the impact of greenery systems, including the mitigation of urban heat islands, the reduction of room temperature, the reduction of internal and external wall surface temperature, and energy saving. The previous investigations are categorized according to the system configuration, green roof, traditional green façade, and double skin green façade.

### 7.1. Discussion 1: Reduction of Urban Heat Index in Green Roof

The urban heat island is a phenomenon where the temperature of the urban area is relatively higher than the surrounding rural areas due to urbanization. The alteration of land surface replaces Earth's natural greenery with tall and dense buildings. Natural greenery is cut down to accommodate the rapid development. Urban building materials are generally non-reflective, impermeable, and have a dark surface, whereby the thermal properties are greater. Hence, greening the building envelope is an ultimate solution to mitigate an urban heat island as a green plant. Urban heat islands contribute to thermal discomfort, as the outdoor temperature is very high.

Table 3 summarizes the previous research conducted for urban heat index in a green roof and is organized according to methodology and the year the paper was published. There have been approximately 18 previous studies conducted on the alleviation of the urban heat island's effect through a green roof, whereby only two of the studies were carried out experimentally and 16 studies were conducted by simulation software. One study was conducted in a tropical climate region, 13 studies were conducted in a subtropical climate region, three studies were conducted in a continental climate

region, and one study had no specific climate region. Most of the research was carried out during the summer season. The maximum reduction of outdoor temperatures around the green roof was between the range of 0.2 °C and 4.2 °C.

**Table 3.** Summary of previous studies on urban heat index in green roof.

| Author & Year | Methodology | Location | Climate Classification | Period of Study | Type of Green Roof | Plant Species | Maximum UHI/Reduction of Outdoor Temp. (°C) |
|---|---|---|---|---|---|---|---|
| Niachou et al., 2001 [79] | Exp. | Loutraki, Greece | Csa | Summer | | | 2 |
| Wong et al., 2003 [30] | Exp. | Singapore | Af | Summer | Intensive | Grass, shrub, and trees | 4.2 |
| Chen et al., 2009 [80] | Simul. | Tokyo, Japan | Cfa | | | | 0.8 |
| Smith and Roebber, 2011 [81] | Simul. | Chicago, USA | Dfa | | Extensive | Grass | 3 |
| Ng et al., 2012 [82] | Simul. | Hong Kong | Cwa | | Intensive and Extensive | Tree (Intensive), Grass (Extensive) | 0.6 |
| Peng and Jim, 2013 [83] | Simul. | Hong Kong | Cwa | Summer | Intensive and Extensive | Tree (Intensive), Grass (Extensive) | 1.7 |
| Ouldboukhitine et al., 2014 [84] | Simul. | France | Cfb | Summer and winter | Extensive | Sedum, grass, herbs | 1 |
| Li et al., 2014 [85] | Simul. | Washington, USA | Csb | | Extensive | Sedum | 1 |
| Chen et al., 2014 [86] | Simul. | Melbourne, Australia | Cfb | | Intensive and Extensive | Woodland, shrub, tussock | 0.5 |
| Lobaccaro and Acero, 2015 [87] | Simul. | Bilbao, Spain | Cfb | | Extensive | Grass | 1 |
| Meek et al., 2015 [88] | Simul. | Melbourne, Australia | Cfb | | Extensive | | 0.9 |
| Sun et al., 2016 [89] | Simul. | Beijing, China | Dwa | | Extensive | Sedum | 2.5 |
| Alcazar et al., 2016 [90] | Simul. | Madrid, Spain | Csa | Summer | Extensive | Sedum and lucerne | 1 |
| Sharma et al., 2016 [91] | Simul. | Chicago, USA | Dfa | Summer | Extensive | Grass | 0.6 |
| Zolch et al., 2016 [92] | Simul. | Germany | Cfb | Summer | Extensive | Grass | 0.5 |
| Berardi, 2016 [93] | Simul. | Toronto, Canada | Cfa | Summer | Extensive | Sedum, mosses, graminaceous | 0.4 |
| Teleghani et al., 2016 [94] | Simul. | California, USA | Csb | | Extensive | Grass | 0.2 |
| Morakinyo et al., 2017 [95] | Simul. | Cairo, Hong Kong, Tokyo, Paris | | Summer | Intensive and Extensive | Tree (Intensive), Grass (Extensive) | 0.6 |

In outdoor experimental research conducted in Singapore, the maximum outdoor temperature around the intensive green roof was 4.2 °C [30]. The experiment was carried out on a roof of a low-rise commercial building and was covered with grass, shrubs, and trees. In this research, six different types of plant species were selected to compare their ability to reduce the temperature. The author concluded that the plant species with a higher leaf area index displayed a lower outdoor temperature due to the dense canopy providing thermal protection.

In simulation research conducted in Chicago, the maximum outdoor temperature around the extensive green roof was 3 °C [81]. The simulation was conducted using the urban canopy model

to study the impact of the green roof on the overall thermal structure in the urban canopy. It was concluded that the roof with green plants had an albedo of 0.9 to 1.0, which was higher than the albedo of other material. A surface with higher albedo can reflect solar radiation and reduce the heat transferred into the building. However, the impact of an actual green roof will be less, as reported in this paper.

### 7.2. Discussion 2: Reduction of Internal and External Wall Temperatures Due to Greenery Systems

The installation of greenery systems, such as a green roof, green wall, traditional green façade, and double skin green façade has improved the thermal performance of the building. The fundamental mechanisms, such as thermal insulation, evapotranspiration, and shading effect, contributed to the improvement of the thermal performance of the building. The thermal performance of the building can be seen in the reduction of internal and external building wall surface temperatures and indoor temperatures. A layer of natural greenery on the building envelope provided an extra layer of insulation and a shading effect. The leaves of the green plant utilized the solar radiation for physiological processes, such as photosynthesis and evapotranspiration. The process of evapotranspiration was the main contributor to the cooling effect of the building, as the leaves of the plant lost water and converted solar radiation into latent heat.

#### 7.2.1. Discussion 2.1: Reduction of Wall Temperature in Traditional Green Façade

Table 4 summarizes the previous research conducted for the reduction of internal and external wall temperatures in a traditional green façade and is organized according to methodology and the year the paper was published. There were around 12 previous studies conducted to study the reduction of wall temperature, whereby six of the studies were case studies, four studies were carried out experimentally, and two were conducted by simulation software. One study was conducted in an arid climate region, eight studies were conducted in a subtropical climate region, and three studies were conducted in a continental climate region. Most of the research was carried out during the summer season. The reduction of internal and external wall surface temperatures was as high as 10 °C.

**Table 4.** Summary of previous studies on reduction of internal and external wall temperature in traditional green façade.

| Author & Year | Methodology | Location | Climate Classification | Period of Study | Plant Species | Reduction of Internal Wall Temperature (°C) | Reduction of External Wall Temperature (°C) |
|---|---|---|---|---|---|---|---|
| Hoyano, 1998 [96] | Case study | Japan | Cfa | Summer | Parthenocissus tricuspidata | 11 | 13 |
| Kohler, 2008 [41] | Case study | Germany | Cfb | Summer, winter | Parthenocissus tricuspidata | | 3 (Summer), 3 (Winter) |
| Sternberg et al., 2011 [97] | Case study | England | Cfb | All year | Hereda helix | | 1.7–9.5 (Summer) |
| Perini et al., 2011 [98] | Case study | Delft, Rotterdam and Benthuizen, Netherlands | Cfb | Autumn | Hereda helix | | 1.2 |
| Yin et al., 2017 [99] | Case study | China | Cfa | Summer | Parthenocissus tricuspidata | | 4.67 (Max.), 2.56 (Average) |
| Cameron et al., 2014 [100] | Exp. | UK | Cfb | Summer | Hereda helix, Stachys byzantina | | 7–7.3 |

**Table 4.** *Cont.*

| Author & Year | Methodology | Location | Climate Classification | Period of Study | Plant Species | Reduction of Internal Wall Temperature (°C) | Reduction of External Wall Temperature (°C) |
|---|---|---|---|---|---|---|---|
| Bolton et al., 2014 [101] | Exp. | UK | Cfb | Winter | Hereda helix | | 0.5 |
| Susorova et al., 2014 [102] | Exp. | USA | Dfa | Summer | Parthenocissus tricuspidata | | 12.6 |
| Elmasry & Haggag [24] | Exp. | UAE | Bwh | Autumn | Plastic bags and various plants | | 1 to 4 |
| Di and Wang, 1999 [103] | Simul. | China | Dwa | Summer | Hereda sp. | | 16 |
| Susorova et al., 2013 [104] | Simul. | USA | Dfa | Summer | Parthenocissus tricuspidata | 2 | 7.9 |

In a case study conducted in a two-story house in Tokyo during summer, the reduction of the internal wall temperature was 11 °C and the external wall temperature was 13 °C [96]. This research was conducted to study how plants control heat transfer into the building and their influence on the thermal performance of the building. This research also highlighted the drawback of convective cooling, as there was stagnant air at night.

In other experimental research conducted in Chicago during summer, the reduction of the external wall temperature was 12.6 °C [102]. The effect of climbing plants on the thermal performance of the building was experimentally studied. The experiment was conducted to measure the thermal performance of four buildings facing North, East, South, and West, respectively. The highest temperature reduction was recorded on the East and West of the building, considering the solar radiation in the morning and evening. It was concluded that the average surface temperature and heat flux was reduced by 10 percent on average.

Other research conducted by simulation showed that the installation of 10 cm thick vegetation during the summer season in Beijing showed a 16 °C reduction of the external wall temperature [103]. In this simulation, there were a few assumptions made, including that the leaves of the plants did not overlap, and the plant had negligible thermal properties. However, the impact on an actual green façade would be less, as reported in this paper.

7.2.2. Discussion 2.2: Reduction of Wall Temperature in Double Skin Green Façade

Table 5 summarizes the previous research conducted for the reduction of internal and external wall temperatures in a double skin green façade and is organized according to methodology and the year the paper was published. There were about eight previous studies conducted to study the reduction of wall temperature, whereby three of the studies were case studies and five studies were carried out experimentally. One study was conducted in a tropical climate region and seven studies were conducted in a subtropical climate region. Most of the research was carried out during the summer season.

Most of the research is usually conducted during the summer season, but this case study was conducted all year-round in Spain. The measurements showed a 15.8 °C reduction of the external wall temperature in summer and an annual average of 5.55 °C [76]. The green façade had a supporting structure of steel and steel sheet to guide the plant. It can be concluded that during the summer and spring season, the green façade had a denser canopy, covering 62 percent of the façade surface. The thermal performance during the summer season was significant because of the evapotranspiration of plants.

**Table 5.** Summary of previous studies on reduction of internal and external wall temperature in double skin green façade.

| Author & Year | Methodology | Location | Climate Classification | Period of Study | Plant Species | Reduction of External Wall Temperature (°C) |
|---|---|---|---|---|---|---|
| Perini et al., 2011 [98] | Case study | Netherlands | Cfb | Autumn | Hereda helix, Vitis, Clemeatis, Jasminum, Pyracantha | 2.7 |
| Perez et al., 2011 [76] | Case study | Spain | Csa | All year | Wisteria sinensis | 15.8 (Max. Summer), 5.55 (Average, all season) |
| Jim, 2015 [105] | Case study | China | Cwa | Summer; Sunny, cloudy, rainy | Ficus pumila, Campsis grandiflora, Bauhinia corymbosa, Pyrostegia venusta | 5 (Sunny), 1–2 (Cloudy), 1–2 (Rainy) |
| Hoyana, 1988 [96] | Exp. | Japan | Cfa | Summer | Dishcloth gourd | 1 to 3 |
| Wong et al., 2010 [106] | Exp. | Singapore | Af | Summer | Climber plants | 4.36 |
| Koyama et al., 2013 [60] | Exp. | Japan | Cfa | Summer | Bitter melon, Morning glory, Sword bean, Kudzu, Apios | 3.7–11.3 |
| Suklje et al., 2013 [107] | Exp. | Slovenia | Cfa/Cfb | Summer | Phaseolus vulgaris "Anellino verde" | 4 |
| Perez et al., 2017 [10] | Exp. | Spain | Csa | Summer | Parthenocissus tricuspidata | 15–16.4 |

In other experimental research conducted by the same author in Spain, but only during the summer season, the external wall temperature was reduced to 15 °C [10]. In their paper, the author focused on the influence of the leaf area index on the thermal performance. This paper also explained how leaf area index was calculated using the direct leaf area index method. It supported the claim that a higher leaf area index reduces the direct solar radiation to the building.

However, Akbari and Kolokotsa [108] reported a review and analysis of the evolution of urban climate change as well as its mitigation technologies over three decades, covering a period of 1985 to 2015. They concluded that the cooling of building interiors can be achieved by two technologies.

- Increasing the solar reflectance to reduce the absorption of solar radiation using materials with high solar reflectance to keep the building surfaces cool. These materials can be used in the building's façade, roofs, and pavements.
- Increasing evapotranspiration in the urban environment, which may be achieved by the intensive use of urban greenery systems, such as parks and green roofs.

*7.3. Discussion 3: Energy Saving and Reduction of Energy Consumption in Building by Greenery Systems*

The building sector is dominating the total energy consumption; the efficient use of energy in buildings is one of the most cost-effective measures to reduce the environmental impact. However, indoor thermal comfort greatly influences the performance of the occupants and is greatly dependent on the operation of heating, ventilation, and air conditioning, resulting in an increase of energy load. Hence, implementing a greenery system to the building envelope is one of the solutions to reduce the energy consumption in the building, as it controls the heat transfer. In other words, a greenery system is a passive technique for energy saving in the building.

### 7.3.1. Discussion 3.1: Energy Saving by Green Roof

Table 6 summarizes the previous research conducted for energy saving in green roofs and is organized according to methodology and the year the paper was published. An approximation of 13 previous studies were conducted to study energy saving, whereby two of the studies were carried out experimentally, and 11 were conducted by simulation software. One study was conducted in a tropical climate region, two studies were conducted in an arid climate region, nine were conducted in a subtropical climate region, and one research was conducted in a continental climate region. Most of the research was carried out during the summer season.

**Table 6.** Summary of previous studies on energy saving by green roof.

| Author | Methodology | Location | Climate Classification | Period of Study | Type of Green Roof | Plant Species | Energy Saving (%) |
|---|---|---|---|---|---|---|---|
| Niachou et al., 2001 [79] | Exp. | Loutraki, Greece | Csa | Summer | | | 2 |
| Wong et al., 2003 [30] | Exp. | Singapore | Af | Summer | Intensive | Grass, shrub, and trees | 0.6–14.5 |
| Saiz et al., 2006 [109] | Simul. | Madrid, Spain | Csa | | Extensive | Sedum, cactus, and desert shrub | 1 |
| Jaffal et al., 2012 [110] | Simul. | La Rochelle, France | Cfb | Summer | Extensive | Sedum | 6 |
| Gagliano et al., 2015 [111] | Simul. | Catania, Southern Italy | Csa | Summer and winter | Extensive | Mosses, sedum, graminaceous, and succulents | 5.1–21.3 |
| Karteris et al., 2015 [112] | Simul. | Thessaloniki, Greece | Csa | | Semi-intensive and extensive | Semi-intensive (Shrub), Extensive (Spices and aromatic plants, herbaceous perennial vegetation, and grasses) | 0.68–6.69 |
| Berardi, 2016 [93] | Simul. | Toronto, Canada | Cfa | Summer | Extensive | Sedum, mosses, graminaceous | 3 |
| Silva et al., 2016 [113] | Simul. | Lisbon, Portugal | Csa | Summer and winter | Intensive, semi-intensive and extensive | Intensive (Moss, sedum, herbaceous and grass) Semi-intensive (Shrub and coppices) Extensive (Tall shrub, large bushes, and trees) | Intensive (45–75) Semi-intensive (10–45) Extensive (25–60) |
| Costanzo et al., 2016 [114] | Simul. | Catania, Southern Italy | Csa | Summer | Extensive | | 10 |
| Mahmoud et al., 2017 [115] | Simul. | Dhahran, Saudi Arabia | Bwh | | Extensive | | 24–35 |
| Foustalieraki et al., 2017 [116] | Simul. | Athens, Greece | Csa | Winter | Extensive | Rosmarinus officinalis, Origanum heraclioticum, Artemisia absinthium, Lavandula dentata, Teucrium fruticans, Lantana Camara, Teucrium marum | 15.1 |
| Khan and Asif, 2017 [117] | Simul. | Riyadh, Saudi Arabia | Bwh | | Extensive | Forbs, sedum, and grass | 6.75 |
| Boafo et al., 2017 [118] | Simul. | Incheon, South Korea | Dwa | | Extensive | | 3.7 |

A numerical experiment conducted using a simulation software in a Mediterranean climate suggested that the green roof had the lowest energy requirement compared to traditional and cool roofs [111], with energy saving of up to 20 percent. In this paper, the author explains that the energy saving in the green roof was due to evapotranspiration and the shading effect of the plant. Another alternative to a green roof is a cool roof. A cool roof has a layer of highly reflective material on the outermost surface of the roof, which has high solar reflectance value to reduce the amount of solar radiation into the building.

Another simulation research conducted in the same Mediterranean climate also suggested that the green roof had a better performance than the cool roof during the summer season, with energy saving of 10 percent [114]. The author also explains that a cool roof increased the energy consumption for heating during the winter season. On the contrary, the green roof did not increase the energy consumption during the winter season due to the shading and insulating properties.

### 7.3.2. Discussion 3.2: Reduction of Energy Consumption by Green Wall

Table 7 summarizes the previous research conducted for energy saving in the green wall and is organized according to methodology and the year the paper was published. There were about six previous studies conducted to study energy saving, whereby one study was conducted using a simulation software, two studies were carried out experimentally, and three of the studies were case studies. One study was conducted in a tropical climate region and five were conducted in subtropical climate regions. Most of the research was carried out during the summer season.

**Table 7.** Summary of previous studies on reduction of energy consumption by green wall.

| Author | Methodology | Location | Climate Classification | Period of Study | Type of Green Wall | Plant Species | Reduction in Energy Consumption (%) |
|---|---|---|---|---|---|---|---|
| Wong et al., 2009 [64] | Simul. | Singapore | Af | | Green wall | | 10%–31% cooling load reduction |
| Cheng et al., 2010 [119] | Exp. | Hong Kong | Cwa | Late summer | Green wall | Zoysia japonica | 30 W/m$^2$ heat flux reduction |
| Chen et al., 2013 [120] | Exp. | Wuhan, China | Cfa | Summer | Green wall | | 2.5 W/m$^2$ heat flux reduction 12% cooling load reduction |
| Mazzali et al., 2013 [50] | Case study | Lonigo, Venice, & Pisa, Italy | Cfa | Summer | Green wall | Several shrubs, herbaceous and climber species | 70 W/m$^2$ heat flux reduction at night 1.5 W/m$^2$ heat flux reduction at night |
| Coma et al., 2017 [45] | Case study | Puigverd de Lleida, Spain | Csa | Winter | Green wall | Rosmarinus officinalis and Helichrysum thianschanicum | 2.96%–4.2% energy saving |
| Perini et al., 2017 [121] | Case study | Genoa, Italy | Cfb | Summer | Green wall | Cistus Jessamine beauty and Cistus crispus | 26.5% energy saving |

A numerical analysis using simulation software comparing three different scenarios in a tropical climate showed that there was a significant 30 percent reduction in energy cooling load with the installation of a vertical greenery system [65]. The author explains that the vertical greenery system greatly reduced the heat transfer through the building façade, resulting in the reduction of mean radiant temperature. The author of [65] also suggested that experimental research should be carried out to compare the results to gain a better understanding of the effects of vertical greenery systems.

A case study conducted in a Mediterranean climate in Italy during summer utilizing the living wall system showed an energy saving of 26.5 percent [121]. The living wall system in this research had

20 different plant species, including both shrubs and climbing plants. In this paper, the temperature of air behind the living wall system was significantly lower than the ambient air temperature. The author explained that the reduction of the temperature led to the decrease of energy consumption required for air conditioning in the room.

## 8. Conclusions

Greenery systems, such as green roofs, traditional green façades, and double-skin green façades, are comprehensively considered and analyzed in the current review. The effectiveness of greenery systems, with evidence from previous research, is also investigated. The results show that greenery systems can improve the thermal performance of buildings with fundamental mechanisms of thermo-fluids and energy conversion, such as thermal insulation, evapotranspiration, and shading effect. A few parameters considerably influence the fundamental mechanisms, such as the leaf area index, foliage height, growing medium, and type of plants. In addition, greenery systems have a few environmental benefits, such as the improvement of stormwater management, air quality, reduction of sound pollution, sequestration of carbon dioxide, and the improvement of building aesthetic.

Literature on the installation of respective greenery system configurations has produced significant results in terms of the reduction of the urban heat index, the reduction of wall surface temperature, and energy conservation. The findings confirm that greenery systems are solutions to urban environment sustainability. Notably, performing a proper comparison of previous studies is difficult because the system configuration, type of plant, climate influence, and other parameters differ. However, the collected and classified data in this work are important in making appropriate decisions on proper greenery systems for building sustainability and building thermal control. Despite drawbacks, the positive results of previous studies suggest the potential of using greenery systems as passive systems for mitigating urban heat islands and reducing the energy load in buildings. Despite the considerable variability of the results obtained, the results still demonstrate a positive impact.

However, it has been realized that a large number of simulation works have contributed to the field and supported many facts on the positive influence of the greenery systems on building sustainability. As such, it is highly recommended to conduct more case studies and experimental investigations to enrich the literature on greenery systems. In particular, experimental studies are advised to evaluate the influence on thermal comfort inside the buildings and the energy saving by greenery systems.

**Author Contributions:** Conceptualization, H.H.A.-K., T.W.B.R., and M.E.; methodology, H.H.A.-K. and K.K.; formal analysis, K.K.; investigation, H.H.A.-K. and K.K.; resources, H.H.A.-K.; data curation, K.K.; writing—original draft preparation, K.K.; writing—review and editing, H.H.A.-K., T.W.B.R., and M.E.; supervision, H.H.A.-K.; project administration, H.H.A.-K., funding acquisition, H.H.A.-K., T.W.B.R., and M.E. All authors have read and agreed to the published version of the manuscript.

**Funding:** This work was carried out under international research collaboration between Universiti Teknologi PETRONAS (UTP)—Malaysia and Universitas Muhammadiyah Surakarta (UMS)—Indonesia under the International Matching Research Grant [CS: 015ME0-054].

**Acknowledgments:** The second authors acknowledge Universiti Teknologi PETRONAS for the financial support under the graduate assistance scheme (GA).

**Conflicts of Interest:** The authors declare no conflict of interest.

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
