# Peer review of "A Comparative Review on Greenery Ecosystems and Their Impacts on Sustainability of Building Environment"

_sustainability, doi:10.3390/su12208529_

Round 1

Reviewer 1 Report

The aim of the work was presentation and review of science papers about  greenery ecosystems and their impact on reduction of urban heat island (UHI) in green roofs, reduction of internal and external wall temperature in green walls, energy saving by green roof and reduction of energy consumption by green wall. Work was based on 120 references.

The authors presented and discussed the key issues and  sorted out the current state of knowledge regarding the urban heat island, the current classification of green systems in the context of the impact on the thermal insulation of the building and energy savings, reviewed and analyzed technological solutions, i.e. green roof and green wall (green facade and double green skin facade), described the basic mechanism of green systems, theirs benefits, plant species and their impact on the analyzed green systems in the context of climate. The authors reviewed   researches publisched in last two decades. The impact of green systems on reducition the urban heat index in a green roof, reduction of wall temperaturę in traditional green facade and in double skin green facade, energy saving (by green roof) or reduction of Energy consumption (by green wall) in  building were analised in the paper. The work is completed with general conclusions proper to content of the paper.

The main value of the paper are analises presentated in tablets 3, 4, 5, 6, 7. Subjects of  the analisies concern important categories of greenery ecosystems and their impact on sustainability of building environment. The authors are aware of the significant variability of the results which are presentation and analysis of chosen  papers effect. Despite of it results demonstrate positive impact and could be useful in making appropriate decisions on proper greenery systems for buildings located especially in tropical climat areas.the leading subject of the analyzes concerns categories that are important for sustainable construction.

The methodology of collecting and selecting the analyzed examples, determining the number of studies analyzed in each group is not clear.  There are different numbers of studies compared in individual analyzes: tab. 3-18 cases, tab. 4-11cases, tab. 5-8 cases, tab. 6 – 13 cases, tab. 7- 6 cases. What was the cause, the principle of it?

There are different numbers of methodology of analised cases (experimental, simulation, case study, …). It is not reflected in conclusion. There is only word „variety”,but it is statment with no valuation of results. F.e. there is lack of experimental or case studies needed to confrontation of simulation results especially in in reduction of urban heat index in green roof cases. It could be useful sugestion for another scientist to continue their researches as verification of symulations.

Significant geographic dispersion (climat zones) makes it difficult to infer, especially in the context of species selection. Conclussion for species selection should be supported by more „in situ” researches than simulation.

Value of this work is that is multidisciplinary subject and its conclusions could be worth for many disciplines scientists.

368- Table 3 – uniformity suggestion – collumn 3 „location” – city, country or just city or just country

461- Table 6-  uniformity suggestion – collumn 3 „location” – city, country or just city or just country

482 -mistake in Table 7:

(1st row/ 6 column) there is „type of green roof” should be „type of green wall”

1st row / 8 column) lack of unit [%]

Author Response

We are so grateful to the time and effort by the review to evaluate and provide valuable comments to improve our article.

It is really appreciated that the respected reviewer captured many drawbacks and recommended valuable comments.  

Reviewer 2 Report

  1. The introduction part is a bit confusing; the authors need to make significant improvements to make it clearer and concentrated. Besides, although the introduction part introduced much basic knowledge in this field, I still not see the gaps (or innovation) of such review researches in this filed. You can read more review papers in this field. Such as Critical reviews on the cooling effect of green infrastructures.

-Taleghani, M. Outdoor thermal comfort by different heat mitigation strategies- A review. Renewable and Sustainable Energy Reviews 81, 2011–2018 (2018).

- Jamie, E., Rajagopalan, P., Seyedmahmoudian, M. & Jamei, Y. Review on the impact of urban geometry and pedestrian level greening on outdoor thermal comfort. Renewable and Sustainable Energy Reviews 54, 1002–1017 (2016).

- Aram, F., Higueras García, E., Solgi, E. & Mansournia, S. Urban green space cooling effect in cities. Heliyon 5, (2019).

- Akbari, H. & Kolokotsa, D. Three decades of urban heat islands and mitigation technologies research. Energy and Buildings 133, 834–842 (2016).

  1. Figure 1 is not a scientific figure, it should redesign.

  1. The discussion part needs to add. In this version, the author did not provide a quality enough discussion, did not focus on the topic you want to discuss. I recommend the author separate these paragraphs as discussion 1, discussion 2, and discussion 3.

  1. The conclusion is not so solid, please reword it.

  1. Please proofread the paper, as there are several writing issues.

Author Response

Dear respected reviewer. We are highly appreciate your time and effort to review our article and suggest vary valuable comments. We have adopted what is meeting with theme of the article and considering you suggestions in the revise version. 

we responded to your comments and we hope that our justifications are satisfactory. 

Thank you so much.

Round 2

Reviewer 2 Report

The introduction part is still lack of sufficient information. When you speak about urban heat island you should mention the mitigation solution. Given that your review concerning green infrastructure, it would be better you explain and categorize green infrastructure including large scale (urban parks, urban foresty, gardens, etc.) and small scale (green roof, green wall, etc.), and then set out the specific subject.

I believe that figure one is not scientific. However, your article is Review but I think it will be published in scientific journals.

The last paragraph of the conclusion did not convince me, It's so common and it doesn't have any novelty. We can see this result in previous studies as well.

Author Response

Dear respected reviewer

We appreciate your time and effort to review our article. Thank you for the comments.

BUT 

  1. The theme of the paper is not on mitigation of urban heat island. The theme of the paper is on the influencing of greenery systems on building. our review is specific and focus. there are many other reviews dealing with the mitigation of Urban Heat Island.
  2. Review paper is not a research paper. Hence, it never expected to have novelty in the review paper. hence, your claim that you cannot find novelty in the last Para in the conclusion is not satisfactory. 

We have responded to your valuable comments in the attached report.

Thank you and stay safe.

Round 3

Reviewer 2 Report

  •  The added Paragraph in the Introduction section needs relevant references.

  •  Figure 1 should be deleted!!

  • The conclusion part needs to make a difference among other review papers in this filed.  

Author Response

Dear respected reviewer.

I am responding to your respected comments based on my experience in publishing more than 280 research, review and technical papers; about 230 of them are indexed in Scopus and around 100 are indexed in ISI.

I am thankful to the time and effort payed by you to improve our article.

My responds to your third round of review comments, are:

  1. You asked to cite a reference to the added statement in the introduction. In fact, we avoid justify and cite general statements and well known facts. accordingly, we discussed and agreed that we lower the value of the paper if we cite general statement.  

  1. You asked to delete figure 1, while you suggested to maintain the figure in your second round of review. The figure is essential to support the paragraph on the description of heat distribution in urban city. 

  1. You ask to make difference in our conclusion. The conclusions are drawn based on the findings of the review and it is totally differ than the previous review papers.

Thank you.